# Higher-Order Multiphoton Absorption Upconversion Lasing Based on ZnO/ZnMgO Multiple Quantum Wells

**DOI:** 10.3390/nano12173073

**Published:** 2022-09-04

**Authors:** Shushu Ma, Haiyuan Wei, Hai Zhu, Francis Chi-Chung Ling, Xianghu Wang, Shichen Su

**Affiliations:** 1School of Semiconductor Science and Technology, South China Normal University, Foshan 528225, China; 2School of Physics, Sun Yat-Sen University, Guangzhou 510275, China; 3Department of Physics, The University of Hong Kong, Hong Kong 999077, China; 4School of Materials, Shanghai Dianji University, Shanghai 200245, China; 5Guangdong Engineering Research Center of Optoelectronic Functional Materials and Devices, Guangzhou 510631, China

**Keywords:** MPA upconversion lasing, localized surface plasmons, multiple quantum wells

## Abstract

In the progress of nonlinear optics, multiphoton absorption (MPA) upconversion lasing enables many vital applications in bioimaging, three-dimensional optical data storage, and photodynamic therapy. Here, efficient four-photon absorption upconversion lasing from the ZnO/ZnMgO multiple quantum wells (MQWs) at room temperature is realized. Moreover, the MPA upconversion lasing and third-harmonic generation peak generated in the MQWs under the excitation of a femtosecond (fs) laser pulse were observed concurrently, and the essential differences between each other were studied comprehensively. Compared with the ZnO film, the upconversion lasing peak of the ZnO/ZnMgO MQWs exhibits a clear blue shift. In addition, the four-photon absorption upconversion photoluminescence (PL) intensity was enhanced in the MQWs/Au nanoparticles (NPs) by the metal-localized surface plasmons (LSPs). The work paves the way for short-wavelength lasers by taking advantage of the high stability and large exciton binding energy of the MQWs’ structures.

## 1. Introduction

It is accepted that nonlinear optics is a fundamental component of modern optics and lies at the core of many classical and quantum technologies. Especially, multiphoton upconversion has versatile applications, such as bioimaging [1], three-dimensional optical data storage [2], and photodynamic therapy [3], due to its considerable advantages of high sensitivity, high resolution, and less photo damage. Multiphoton absorption (MPA) upconversion is the process that two or more low-energy photons are simultaneously absorbed and converted into one photon of higher energy by the anti-Stokes effect under a high excitation density. Compared to traditional nonlinear crystals [4] (e.g., BBO or KDP) that use second- and third-order harmonic generation to achieve short-wavelength lasers, strict phase matching and polarization orientation are not necessary for multiphoton-excited lasers [5]. Hence, the tuning range of pumping light for the MPA upconversion laser can be extended significantly.

Zheng et al. demonstrated an efficient upconverted stimulated emission induced by four-photon and five-photon absorption in an organic solution [6]. Five-photon-excited upconversion fluorescence has also been demonstrated in the family of halide perovskite colloidal nanocrystals [7]. In addition, three-photon- and four-photon-excited red/near-infrared fluorescence of the carbon nanodots have also been observed [1]. Nevertheless, the intrinsic liquid properties of the organic solution and the poor antiradiation ability of perovskite nanocrystals are the bottleneck of practical applications. Semiconductor crystals may be more suitable for such applications because of their large damage resistance threshold and high optical gain. ZnO is often considered to be a possible candidate for upconversion material due to its wide and direct bandgap (3.37 eV), large exciton binding energy (60 meV), low production costs, and excellent optical properties.

Although five- and six-photon absorption upconversion lasing have been achieved in ZnO-based semiconductors [8,9], to the best of our knowledge, there are relatively few studies on the multiphoton-excited lasing of ZnO/ZnMgO multiple quantum wells (MQWs). The excitons in the quantum well structure have higher stability and exciton binding energy. The physical mechanism is that the quantum well structure effectively confines the carriers in a two-dimensional space, so that the carrier wave functions in the well layer overlap, thereby increasing the exciton binding energy. To obtain stronger MPA upconversion photoluminescence (PL) intensity, the metal-localized plasmon effect was studied. Surface plasmons have been extensively used to enhance the luminescence intensity of ZnO-based semiconductor materials owing to the surface localized fields and resonant coupling effects. In 2004, Okamoto et al. described a method to enhance the efficiency through the energy transfer between quantum wells and surface plasmons [10]. Yeh et al. demonstrated the PL intensity enhancement by depositing Ag nanoparticles (NPs) on the surface of the quantum wells [11]. Zhang et al. reported the surface-plasmon-enhanced n-ZnO/AlN/p-GaN light-emitting diodes by inserting the Ag NPs between the ZnO and AlN layers [12]. Concerning the use of metal localized surface plasmons (LSPs) to improve the band-edge luminous efficiency of ZnO-based devices, Liu’s group has conducted a lot of research [13,14,15,16,17]. The use of metal NPs including Al, Ag, and Au, etc., was reported in enhancing the luminous intensity of ZnO-based systems [18,19,20].

The current paper demonstrates the MPA upconversion PL in the ZnO/ZnMgO MQWs. A femtosecond (fs) pulse was employed to study the four-photon absorption upconversion lasing in the ZnO/ZnMgO MQWs. The observed coexisting dominant emission peak and third-harmonic generation peak confirmed the lasing action in the MQWs derived from the high-order MPA process. In addition, the MPA upconversion PL intensity of the MQWs was enhanced by coating the surface with gold nanoparticles. The current results validate the feasibility of high-order MPA upconversion lasing in MQWs and the enhancement in upconversion PL intensity by metal LSPs.

## 2. Materials and Methods

### 2.1. Fabrication

The gain media (ZnO/ZnMgO MQWs) with a highly preferred orientation were fabricated on sapphire (Al_2_O_3_) substrates by the plasma-assisted molecular beam epitaxy (PA-MBE) [21]. For the growth of the gain media, the substrate was loaded into an ultrahigh-vacuum chamber and annealed at 800 °C for 30 min to remove the surface contaminants. Elemental Zn (99.9999%) and Mg (99.999%) were evaporated via conventional effusion cells. Pure oxygen (99.999%) was used as the oxygen source, and the oxygen plasma was generated by radio frequency (rf)-activated radical cell. In the experiment, the rf power of oxygen plasma was 300 W. A high-quality ZnO buffer layer was grown under the temperature of 800 °C. Then, the ZnO/ZnMgO MQWs were prepared in the growth pressure of 1.0 × 10^−6^ mbar at 600 °C. The designed structure of ZnO/ZnMgO MQWs is depicted in Figure 1a. As shown, the ZnO well and ZnMgO barrier were repeatedly grown in ten periods. The thickness of ZnO well and ZnMgO barrier was 6 nm and 8 nm, respectively. Here, the composition of Mg was about 15% in ZnMgO barrier.

In order to fabricate the disordered metal nanostructures, the Au ultrathin monolayer was deposited on the upper surface of the ZnO/ZnMgO MQWs by thermal evaporator at the pressure of 1.0 × 10^−6^ Torr and a deposition rate of 0.4 Å s^−1^ [22]. For comparison, half of the upper surface was kept without any process. Subsequently, the Au ultrathin monolayer (5 nm thickness) was annealed for 2 h under argon condition at 550 °C. Consequently, half part of the upper surface was deposited with Au nanoparticles.

### 2.2. Characterization and Measurements

The detailed microscopic structure of the ZnO/ZnMgO MQWs was characterized by field-emission scanning electron microscopy (SEM). The X-ray diffraction (XRD) patterns were obtained with an X-ray diffractometer. The temperature-dependent PL spectra of the MQWs were measured in a temperature range from 77 K to room temperature (RT) with a He–Cd laser (325 nm) as the excitation source. A home-built PL measurement installation was used to investigate the optical properties of the MQWs. The tunable femtosecond laser pulses of 50 fs duration were generated from the optical parametric amplifier (OPA) system. The latter was pumped by a regenerative amplified mode-locked Ti: sapphire oscillator at a center wavelength of 800 nm at 1 kHz repetition rate. The pumping light is focused on the sample through the objective lens. The upconverted emission signal is also collected by the same lens and coupled into a spectrometer. The corresponding power was measured by an optical power meter.

## 3. Results and Discussion

The sample was characterized by SEM and XRD. The SEM image of the sample in Figure 1b indicates a smooth surface. The diffraction pattern is shown in Figure 1c. The X-ray diffraction in Figure 1c exhibits the diffraction peak at 34.44°, which is attributed to the (002) plane of the ZnO hexagonal crystal having the c-axis orientation. The full width at half-maximum (FWHM) of the (002) peak is 0.06°. Figure 1d shows the X-ray rocking curve (XRC) of the ZnMgO sample, with the FWHM of 0.3°, indicating that the sample has good crystal orientation and high crystal integrity.

The PL spectra taken at different temperatures ranging from 77 K to 300 K are shown in Figure 2a. Remarkably, as the temperature changes from 77 K to 300 K, the emission of the MQWs exhibits an obvious redshift from 3.402 eV to 3.341 eV. Meanwhile, its linewidth widens with rising temperature. It is worth noting that efficient excitonic emission from the MQWs can also be noticed at room temperature (Appendix A). Furthermore, the temperature-dependent photon energy was also investigated. The photon energy corresponds to the position where the PL peak intensity is strongest (Appendix A). In Figure 2b, the PL peak position shifts progressively to the low-energy side at about 61 meV with the increase in temperature. The variation of the bandgap of MQWs with temperature is given by the Varshni empirical formula [23]:E_g_ = E_0_ − αT^2^/(β + T)(1)
where E_g_ is the energy gap, which may be direct (E_gd_) or indirect (E_gi_), E_0_ is the bandgap at 0 K, and α and β are the constants. The red solid line in Figure 2b is the fitting curve to the experimental data points through the Varshni relation, with the fitted values of: E_0_ = 3.41 eV, β = 700 K, and α = 6.64 × 10^−4^ eV/K. In addition, the temperature dependence of the PL spectra from the ZnO film also shows similar behavior (Appendix A). The fitting results of E_0_ = 3.38 eV, β = 710 K, and α = 8.66 × 10^−4^ eV/K are in good agreement with those of ZnO reported by L. J. Wang et al. [24]. We find that the value of E_0_ from the MQWs is larger than the ZnO film. The blueshift is due to the distinct quantum confinement effect.

The four-photon upconversion lasing in a semiconductor is schematically illustrated in Figure 3a. The electron in the valence band (VB) can be excited into the conduction band (CB) using virtual energy levels via simultaneous absorption of several low-energy photons that are smaller than the energy of bandgap. Subsequently, the electrons in the conduction band will transit into the valence band and emit photons of shorter wavelengths. Under a high excitation density, the population inversion of carriers will be formed in the media, and then an amplified stimulated emission can be achieved [8]. Meanwhile, harmonic generation processes caused by the virtual energy levels may also be likely to occur and emit photons with energies below or above the bandgap. The observation of harmonic generation can be considered as the supporting evidence for the lasing resulting from MPA upconversion [25].

The optical properties of the MQWs are investigated carefully via a home-built PL measurement installation, as shown in Figure 3b. A tunable fs-pulsed laser (1550 nm) is used as the excitation source. The laser is focused on the sample through the objective lens. The upconverted emission signal is still collected by the same lens and coupled to the spectrometer. At high pumping density, a typical complete emission spectrum including the dominant emission peak (centered around 389 nm) and the third-harmonic generation (THG) peak (centered around 508 nm) from the sample at RT is given in Figure 4a. Here, all spectra have been normalized. The dominant emission peak corresponds to the intrinsic near-band-edge emission (NBE) of the MQWs. It is worth noting that the NBE peak is associated with the MPA upconversion unambiguously [9].

Further power-dependent MPA upconversion PL study under the same experimental conditions was also performed. As illustrated in Figure 4b, a narrow peak emerges, and its PL intensity rises with the increase of excitation power density. Meanwhile, the dominant emission peaks display a slight redshift as the excitation power density increases to 1.214 MW/cm^2^. In the experiments, we also realized upconversion PL lasing based on ZnO thin films by the same excitation light. Figure 4c shows the lasing peak of the ZnO/ZnMgO MQWs exhibiting a clear blue shift and the central wavelength of the peak shifting from 393 nm to 388 nm, while the spectrum for the ZnO thin films is included for comparison. This phenomenon is consistent with the blueshift of the corresponding emission peak mentioned above.

Furthermore, the integrated emission intensity and FWHM of the MQWs upconversion PL lasing (the dominant emission peak) versus the excitation power density are given in Figure 4d. At pump energy slightly lower than the value of the kink point (lasing threshold), the PL intensity increases slowly, and a broad spontaneous emission (SPE) band centered at 389 nm corresponds to the intrinsic NBE of the MQWs. When the pump energy is larger than the lasing threshold, a higher slope efficiency is observed, whereas the FWHM of the NBE peak reduces suddenly from 18.7 nm to 9 nm. Here, the kink point indicates that the SPE from the sample will convert into a stimulated emission. This is a vital performance of MPA-amplified spontaneous emission (ASE) [8].

Given that the bandgap of MQWs (3.41 eV is given above) is greater than the photon energy of the excitation light (0.8 eV), the super-linear behavior of the narrow peak stemming from the stimulated ASE is realized via MPA simultaneously. The peak of ASE shifts to a long wavelength slightly with the excitation density increasing because of bandgap shrinkage caused by a thermal effect. However, the peak of THG is not influenced by the external environment (Appendix A). Moreover, Figure 4e indicates that the FWHM of THG has no obvious change, except that the emission intensity gradually increases with increasing power density. Indeed, the THG peak is associated with the tuning of the pump light because of the fundamental frequency light shift [5]. The above phenomena show the essential distinction between the dominant emission peak and THG peak and confirm that the ASE produced in the MQWs is the result of the radiative transition process of MPA upconversion.

To obtain stronger upconversion lasing, the metal-localized plasmon effect was studied. Metal surface plasmons are surface electromagnetic waves propagating at the interface between the metal and medium, interacting with free electrons and resulting in the collective oscillation of free electrons at the interface between the metal and medium. For relatively small metal particles, the metal surface plasmons are localized near the particles, which are called LSPs. In recent years, many scholars have investigated various ZnO structures with metal NPs. For example, Wei et al. reported a one-dimensional microbelt/Au-nanoparticles construction to enhance the nonlinear third harmonic emission via the LSP effect [26]. Wang et al. reported enhanced UV emission from ZnO on Ag NP arrays by the LSP effect [27]. To date, the enhancement of high-order multiphoton upconversion UV lasers by localized surface plasmon resonance effect has been rarely reported and requires in-depth analysis.

Using metal LSPs to enhance the NBE emission peak of ZnO/ZnMgO MQWs, the key is to adjust the resonance frequency of the metal LSPs to match the NBE peak of the MQWs. The resonance frequency of the metal LSPs is related to many factors, such as the shape of the NPs, size and surrounding dielectrics, etc. Furthermore, the resonant absorption peak of the plasmon will be red-shifted with the increasing size of the NPs [22]. The enhancement effect not only depends on the matching degree of the metal surface plasmon resonance energy and the luminescence energy but also on the scattering ability of metal NPs [28,29]. Therefore, it is very necessary to prepare metal particles with a strong scattering ability to enhance the luminescence of materials.

To study the properties of light–matter nonlinear interaction from the ZnO/ZnMgO MQWs/Au NPs, the Au ultrathin monolayer was first deposited on the upper surface of the ZnO/ZnMgO MQWs in a vacuum thermal evaporation chamber [22]. In order to fabricate the disordered Au NPs, the Au ultrathin monolayer was processed by the thermal annealing method [12]. More details about the preparation process are described in the experimental section. The schematic of the ZnO/ZnMgO MQWs/Au NPs composite is shown in Figure 5a. For comparison, half part of the upper surface was deposited with metal NPs, and the other half was not. For our sample, the local morphology and density of Au NPs are random in the inset of Figure 5a, while the average size is macroscopically controllable, and the average size of randomly distribute Au NPs is 25 nm (Appendix A). It can be considered to enhance upconversion PL intensity due to the part of Au NPs generating a plasmon resonance peak that almost matches the ZnO/ZnMgO MQWs PL band peaked at ∼389 nm. In addition, after the formation of NPs, the scattering efficiency of NPs is greatly improved compared with metal films, which is also one of the reasons for the enhanced luminescence.

We investigated the enhancement of the MPA upconversion PL by the measurement installation mentioned above. The PL spectra of the ZnO/ZnMgO MQWs/Au NPs were measured under the same incident power at RT. The PL spectra of the sample covered with and without Au NPs are presented in Figure 5b. It can be seen clearly that the NBE emission intensity from the MQWs/Au NPs is almost two times larger than the MQWs sample. The MPA upconversion PL enhancement of the MQWs/Au NPs was achieved by metal LSPs.

## 4. Conclusions

In summary, we have demonstrated four-photon absorption upconversion PL lasing from the near-infrared to ultraviolet region in a novel gain medium (ZnO/ZnMgO MQWs). Furthermore, the coexistence of a dominant emission peak and THG peak was observed in the MQWs under the excitation of a high-power fs laser pulse, where the THG peak is related to the tuning of the pump light, and the dominant emission peak is the result of the radiative transition process of MPA upconversion. In contrast to the ZnO film, the upconversion PL lasing peak of the ZnO/ZnMgO MQWs exhibits a clear blue shift from 393 nm to 388 nm. Finally, based on the metal LSPs, the properties of light–matter nonlinear interaction from the ZnO/ZnMgO MQWs/Au NPs were explored. As a result, the high-order MPA upconversion PL lasing was enhanced twofold relative to the bare MQWs.

## Figures and Tables

**Figure 1 nanomaterials-12-03073-f001:**
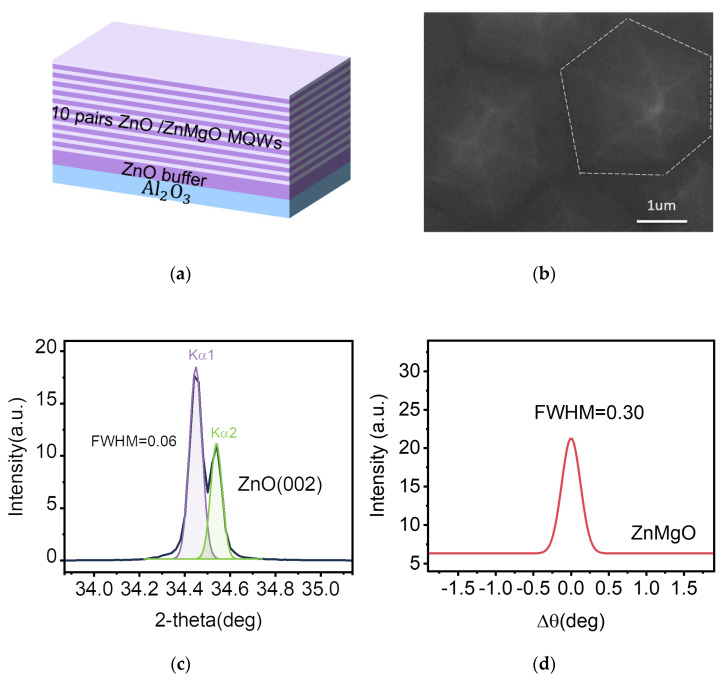
Structure characterizations of the ZnO/ZnMgO MQWs. (**a**) Schematic of the ZnO/ZnMgO MQWs. (**b**) The SEM image of the surface morphology of ZnO sample. (**c**) The XRD pattern of the ZnO sample. (**d**) The XRC of the ZnMgO sample.

**Figure 2 nanomaterials-12-03073-f002:**
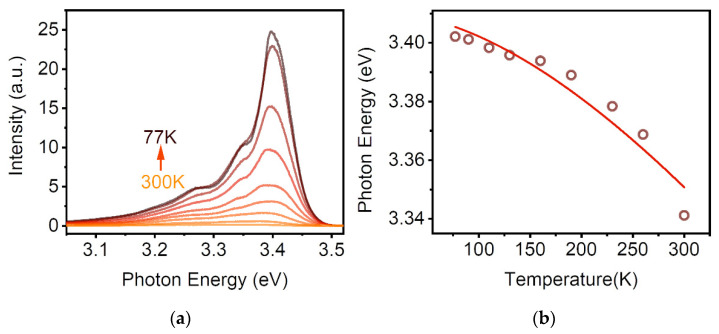
The temperature dependence of optical properties for the ZnO/ZnMgO MQWs. (**a**) Temperature-dependent PL spectra of the MQWs from 77 K to 300 K. (**b**) PL peak position with temperature for the MQWs. Red solid lines show the fits to the experimental data.

**Figure 3 nanomaterials-12-03073-f003:**
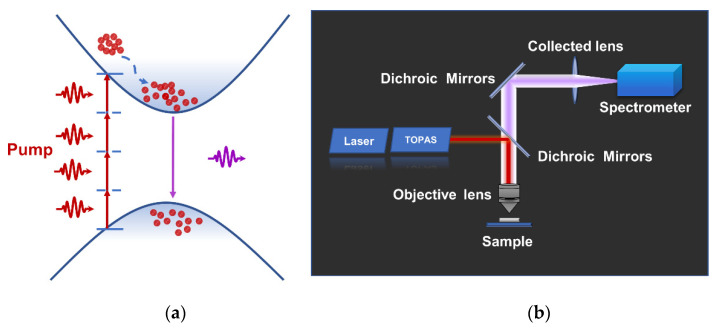
Schematic diagram for MPA upconversion lasing and device schematic. (**a**) Energy diagram indicating the mechanism for upconversion transition in virtual energy levels and bandgap by simultaneous MPA. (**b**) Setup of the MPA (schematic) for the MQWs based on a tunable fs-pulsed laser.

**Figure 4 nanomaterials-12-03073-f004:**
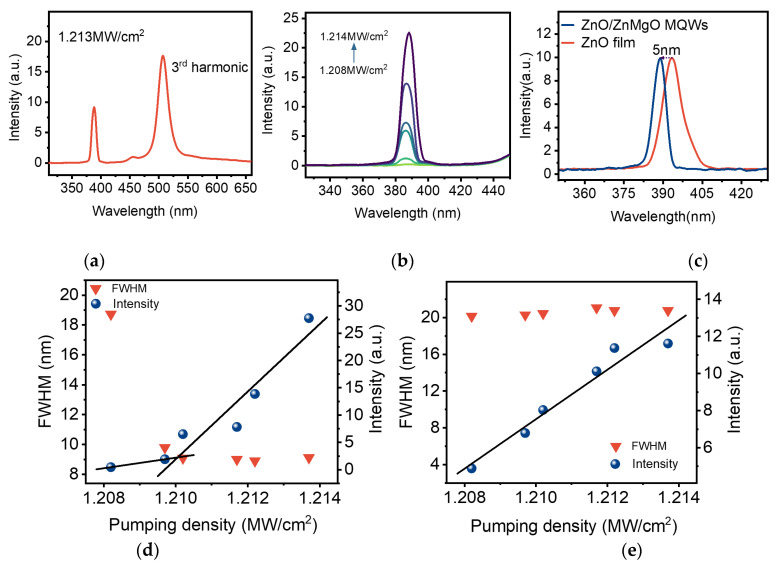
Analysis of MPA upconversion optical characteristics. (**a**) A typical complete PL spectrum includes the dominant emission peak and the third harmonic peak from the MQWs at room temperature. (**b**) The spectrum of dominant emission peaks in the MQWs under different excitation power. (**c**) The upconversion PL lasing spectra of the ZnO films and ZnO/ZnMgO MQWs at RT. (**d**) The experimental data of dominant emission peak in MQWs for the integrated emission intensity and FWHM versus the excitation power show a “kink”. (**e**) The integrated emission intensity and FWHM of the third harmonic peak in MQWs versus pumping power.

**Figure 5 nanomaterials-12-03073-f005:**
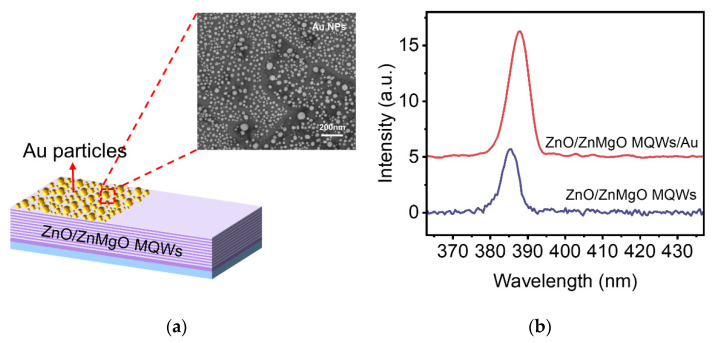
The Schematic and MPA upconversion PL spectra of ZnO/ZnMgO MQWs/Au NPs. (**a**) Schematic of ZnO/ZnMgO MQWs/Au NPs composite. Inset, the SEM image of Au NPs on ZnO/ZnMgO MQWs. (**b**) The MPA upconversion PL spectra of dominant emission peaks in the ZnO/ZnMgO MQWs covered with and without Au NPs.

## Data Availability

Not applicable.

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
