# Peer review of "Higher-Order Multiphoton Absorption Upconversion Lasing Based on ZnO/ZnMgO Multiple Quantum Wells"

_nanomaterials, 2022, doi:10.3390/nano12173073_

Round 1
Reviewer 1 Report
The work of Shushu Ma et al. presents multiphoton absorption in ZnO/ZnMgO quantum wells and additionally shows influence of gold nanoparticles of optical properties of examined QW. Paper is well organized, however the current version have to be corrected before the publication. Below I listed my remarks:
- for citation use et al. instead of et al (see lines 41, 61 ...)
- line 81: acronym for plasma assisted molecular beam epitaxy is PA-MBE, not P-MBE
- authors used quite strange symbol for Celsius grade, see lines 83 and 96 for instance
-2.1 Fabrication - I would add information about 10 repetition of QW in the text
- How authors determined the Mg content in barrier layer?
- Figure 1a - please indicate both components of XRD signal presented in this figure
- lines 149-152: authors mentioned that low energy absorption is performed via virtual energy levels. Are this levels somehow related with defects occurring in ZnO/ZnMgO structure?
- general issue - please place Figures as close as possible just after first referring in the text
- please separate numbers and unit with space, see lines 179 vs 183
- Figure 4: figure 4b: please recalculate laser power to power density; figure 4c: please indicate/mark the shift in the graph; figure 4d and 4e add information what is the meaning of dot and triangle
- Figure 5 - do you have histogram of size distribution of Au NPs?
- lines 262-268 - for me authors should perform similar investigations as shown in Figure 4 for samples decorated with gold nanoparticles
Reviewer 2 Report
11) Figure 1(c) - Why is such a short section of the XRD spectrum shown? how do the authors explain why the maximum is a doublet?
22) Figure 2(a) – it is claimed that “As depicted in Figure 2a, efficient excitonic emission from the MQWs can be noticed at room temperature” – actually, it is impossible to see the spectrum at temperatures close to room temperature
33) Figure 2(b) - The photoluminescence spectrum consists of many components. What component was taken into account in the analysis and how was the spectrum approximation done? Separate spectra at each temperature with their approximation should be presented in supplementary materials. In addition, how is the complex shape of the photoluminescence spectrum explained?
44) The main conclusions of the work are based on the spectra shown in Figure 4(b). To be more precise, the observed “kink” in the dependences of the PL intensity and the spectrum width is associated exclusively with one point, 23.71 μW. However, at this power, the spectrum is practically indistinguishable. With this signal level, any intensity and spectrum width can be calculated. In addition, the studied range of laser excitation power variation is less than 0.5% of the maximum power, which is generally much lower than the typical power stability of these laser systems. It is also not clear how accurately this power can be measured. It is not clear why such a small power range is shown, because such studies usually consider changes in the power of the exciting radiation by several orders of magnitude in order to observe different luminescence regimes.
55) Line 248 – “For our sample, the average size of randomly distribute Au nanoparticles are 25 nm in the inset of Figure 5a. It can be considered to enhance upconversion PL intensity due to the part of Au NPs generating a plasmon resonance peak that almost matches the ZnO/ZnMgO MQWs PL band peaked at ∼ 389nm.”
First, there are no Au nanoparticle size estimation results provided. Secondly, no results are presented regarding the plasmon resonance band in the formed nanoparticles.
66) It is claimed that the use of Au nanoparticles aims “To obtain stronger upconversion lasing”. However, the authors provided the only 1 spectrum at the particular (and unknown) excitation power. For this section, the dependencies of FWHM and integrated PL intensities for both sides of the sample must be provided first (in a wide excitation power density range), and quantitative comparison must be made then.
Round 2
Reviewer 1 Report
I am satisfied with authors response and correction.
Author Response
Dear editor and reviewers
We sincerely thank the editor and reviewers for their enthusiastic work. Thanks again for the reviewers’ professional evaluations, which have further improved our manuscript.
If you have further requirement, please do not hesitate to let me know.
Looking forward to hearing from you.
Thanks for your consideration.
Yours sincerely,
Shichen Su
Reviewer 2 Report
The authors made some improvements and addressed several remarks. However, I still afraid that the main statements of the study are not supported by the reliable experimental data.
The variation of the power density is 0.5% that is comparable with the accuracy of the used optical power meter (+-0.5 f.s.). The stability of the laser system is not specified. The errors in the calculation of FWHM and intensity are not indicated. Despite the authors claim that they made “many experiments”, there is no statistical analysis. Interestingly, even the specified optical power meter has a built-in function of a statistical analysis. That’s why I would say that the main conclusion about lasing is based on the one indistinguishable spectrum.
Another conclusion about the LSPR-induced enhancement of the lasing is not supported by the corresponding power-dependent measurements.
That’s why I believe that the manuscript can not be published without substantiated evidences of the observation of this phenomenon.
